# Sex Differences and Predictors of In-Hospital Mortality among Patients with COVID-19: Results from the ANCOHVID Multicentre Study

**DOI:** 10.3390/ijerph18179018

**Published:** 2021-08-26

**Authors:** Nicolás Francisco Fernández-Martínez, Rocío Ortiz-González-Serna, Álvaro Serrano-Ortiz, Mario Rivera-Izquierdo, Rafael Ruiz-Montero, Marina Pérez-Contreras, Inmaculada Guerrero-Fernández de Alba, Álvaro Romero-Duarte, Inmaculada Salcedo-Leal

**Affiliations:** 1Unidad de Gestión Clínica Interniveles de Prevención, Promoción y Vigilancia de la Salud, Hospital Universitario Reina Sofía, 14004 Córdoba, Spain; fernandez.martinez.nf@gmail.com (N.F.F.-M.); rocio.ortiz.gonzalezserna.sspa@juntadeandalucia.es (R.O.-G.-S.); alvaro.serrano.sspa@juntadeandalucia.es (Á.S.-O.); minmaculada.salcedo.sspa@juntadeandalucia.es (I.S.-L.); 2Preventive Medicine and Public Health Research Group, Maimonides Institute for Research in Biomedicine of Cordoba (IMIBIC), 14004 Córdoba, Spain; 3Service of Preventive Medicine and Public Health, Hospital Universitario Clínico San Cecilio, 18016 Granada, Spain; marioriveraizquierdo2@gmail.com (M.R.-I.); inmagfa@gmail.com (I.G.-F.d.A.); 4Instituto de Investigación Biosanitaria, ibs. Granada, 18012 Granada, Spain; 5Department of Preventive Medicine and Public Health, University of Granada, 18016 Granada, Spain; 6Facultad de Medicina y Enfermería, Universidad de Córdoba, 14004 Córdoba, Spain; 7Service of Preventive Medicine and Public Health, Hospital Universitario de Puerto Real, 11510 Puerto Real, Spain; marinaperezcon@gmail.com; 8Service of Preventive Medicine and Public Health, Complejo Hospitalario de Jaén, 23007 Jaén, Spain; 9School of Medicine, University of Granada, 18016 Granada, Spain; rufete@correo.ugr.es

**Keywords:** COVID-19, SARS-CoV-2, hospital mortality, risk factors, sex

## Abstract

Spain is one of the countries most affected by the COVID-19 pandemic. Although risk factors for severe disease are published, sex differences have been widely neglected. In this multicentre study, we aimed to identify predictors of in-hospital mortality in men and women hospitalised with COVID-19. An observational longitudinal study was conducted in the cohort of patients admitted to four hospitals in Andalusia, Spain, from 1 March 2020 to 15 April 2020. Sociodemographic and clinical data were collected from hospital records. The Kaplan–Meier method was used to estimate 30-day survival and multiple Cox regression models were applied. All analyses were stratified by sex. A total of 968 patients were included (54.8% men, median age 67.0 years). In-hospital mortality reached 19.1% in men and 16.0% in women. Factors independently associated with an increased hazard of death were advanced age, higher CURB-65 score and not receiving azithromycin treatment, in both sexes; active cancer and autoimmune disease, in men; cardiovascular disease and chronic lung disease, in women. Disease outcomes and predictors of death differed between sexes. In-hospital mortality was higher in men, but the long-term effects of COVID-19 merit further research. The sex-differential impact of the pandemic should be addressed in public health policies.

## 1. Introduction

Since its discovery, severe acute respiratory syndrome coronavirus 2 (SARS-CoV-2) has quickly spread across the globe causing devastating effects. With more than 3.8 million confirmed cases and 80,000 reported deaths as of 30 June 2021 [1], the burden of disease attributable to COVID-19 in Spain is among the highest in the world, not to mention the indirect effects on other diseases. Coronavirus hit hardest in the first wave of the pandemic, characterised by the lack of preparedness from a public health standpoint, the use of drugs with insufficient evidence and, above all, an unprecedented health care collapse.

Several studies have described the clinical spectrum of COVID-19. It ranges from asymptomatic to severe acute respiratory syndrome (SARS) with a variety of complications and, in worst cases, death. Such variability is not random: patients with obesity, chronic conditions or malignancies, those aged > 60 years and immunocompromised hosts, among others, are at higher risk of death [2]. Although many risk factors have been identified, their contribution may not be the same in women and men. Ignoring sex differences in COVID-19 mortality hampers our understanding of infection by SARS-CoV-2 and, indirectly, neglects the importance of gender-based risk factors [3]. For example, previous works have suggested that factors associated with COVID-19 outcomes are different in men and women [4]. However, further research is warranted and several authors have pointed to the need of studies stratified by sex [5].

Furthermore, geographical singularities must be considered. In the context of the first epidemic wave, disparities among populations were shaped not only by their demography. Social determinants of health [6]—material conditions, psychosocial circumstances and behavioural factors—influenced the risk of infection among the exposed, while health systems (overload and changing protocols for diagnosis, admission and treatment) influenced the risk of death among the infected. As a result, important differences were observed across areas even in the same country [7]. To our knowledge, risk factors for in-hospital mortality have been published so far from four Spanish regions [7,8,9,10].

Andalusia is the most populated region in Spain (8.4 million inhabitants). Despite not being among the most affected regions in the first wave, it had up to 12,568 confirmed cases as of 10 May 2020, of which 6209 were hospitalised and 1444 died [11]. Besides, limited availability of resources—annual per capita expenditure on health care is the lowest in the country (EUR 1262) [12]—has compromised the healthcare’s response to COVID-19. In this multicentre study, our objectives were to describe the baseline characteristics and to analyse the predictors of in-hospital mortality of adult men and women admitted to four Andalusian public hospitals during the first wave of the pandemic.

## 2. Materials and Methods

### 2.1. Design and Setting

We conducted a retrospective cohort study, according to the STROBE guidelines [13] (see Appendix A). The Andalusian Cohort of Hospitalized patients for COVID-19 (ANCOHVID) [14] is a dynamic cohort of patients admitted to four Andalusian public hospitals—Reina Sofía University Hospital (RSUH, Córdoba, Spain), San Cecilio University Hospital (SCUH, Granada, Spain), Ciudad de Jaén University Hospital (CJUH, Jaén, Spain) and Puerto Real University Hospital (PRUH, Cádiz, Spain)—from 1 March 2020 to 15 April 2020. All patients were followed since admission until death, discharge, or six months after the date of first admission if they were still hospitalised. Therefore, the date of end of follow-up was 15 October 2020. RSUH, SCUH and CJUH cover, altogether, a population of one million people in large urban areas, while PRUH covers a population of approximately 150,000 in mostly rural areas. Average life expectancy in the region was 82 years before the pandemic, with a 5-year gap between men (80) and women (85). These public hospitals contain a large number of beds (RSUH, 1209; CJUH, 749; SCUH, 543; PRUH, 319) and are part of the Spanish national health system, which provides universal coverage to any of its residents.

Inclusion criteria of the study were: confirmed polymerase chain reaction (PCR) to SARS-CoV-2 of nasal, pharyngeal, sputum or bronchoalveolar lavage sample, following the Spanish Ministry of Health’s definition of confirmed cases [15] as of 1 March 2020, at the beginning of the inclusion period; and age at diagnosis of 18 years or older.

### 2.2. Variables and Data Sources

We collected clinical data from hospital electronic health records. Sociodemographic data were retrieved from primary care electronic health records. We then merged both types of data in a pre-defined institutional database.

The variables included in this study were: (a) sociodemographic characteristics: age (years), sex and country of birth (native vs. non-native), place of residence and dependency in activities of daily living (ADL); (b) chronic conditions: arterial hypertension, diabetes, cardiovascular disease, chronic lung disease, chronic obstructive pulmonary disease (COPD), non-allergic asthma (hereafter, asthma), chronic kidney disease, active cancer (solid or hematologic malignancy), history of cancer in the previous 5 years, autoimmune disease, obesity, tobacco smoking, history of previous transplantation, and HIV infection; (c) treatments received: polymedication (i.e., six or more prescribed drugs) prior to admission, immunosuppressive therapy prior to admission, antimicrobial agents (hydroxychloroquine, lopinavir-ritonavir, azithromycin and other antibiotics), immunosuppressants (tocilizumab and high-dose systemic corticosteroids); (d) clinical: abnormal chest X-ray at admission, ferritin levels upon admission, acute distress respiratory syndrome (ADRS), concurrent infection (coinfection or superinfection during hospitalisation), need for mechanical ventilation (invasive—IMV and non-invasive—NIMV), length of hospital stay and length of intensive care unit ICU) stay, and the CURB-65 severity score at the time of hospital admission; (e) outcomes: admission to the ICU and death during hospital stay (main outcome). We considered the main outcome for patients who died during the first admission and for those readmitted due to COVID-19 who died during readmission. Obesity, smoking and ADRS were excluded from the analyses because data were missing in ≥25% of patients. Definitions of variables are presented in Appendix A.

### 2.3. Statistical Analyses

Qualitative variables were expressed as absolute numbers and their relative frequencies. Normality of quantitative variables was assessed by the Shapiro-Wilk test, and they were expressed as mean and standard deviation (SD) if normally distributed, or as median and interquartile range (IQR) if not. In the bivariate analysis, we compared categorical variables with Pearson’s chi-square test or Fisher’s exact test, when appropriate; and quantitative variables with Student’s *t*-test or Mann-Whitney U test, when appropriate. To further characterise the profile of comorbidities among men and women who died during hospital stay, we built correlation matrices to examine interdependencies between the main chronic conditions in patients who had at least one of them.

We used the Kaplan-Meier method to estimate the probability of survival and the log-rank test to compare survival between groups. In the multivariate analysis, all variables associated with in-hospital mortality in the bivariate analysis with a *p*-value < 0.30 and key comorbidities were included in two backward, multiple Cox regression models (one per sex). Models were adjusted by age, ADL, chronic conditions, treatments received, concurrent infection, need for mechanical ventilation and CURB-65 score. For each variable, we used Schoenfeld residuals to verify the proportionality of hazards. Collinearity of the model was measured by the variance inflation factors. Significance of the regression coefficients was assessed by Wald’s test. All statistical tests were two-sided, and variables with a *p*-value of 0.05 or less were considered significant. The analyses were performed using R software version 4.0.3 (R Foundation for Statistical Computing, Vienna, Austria).

### 2.4. Ethical Considerations

The procedures described here were carried out in accordance with the ethical standards described in the Helsinki Declaration revised in 2013. Ethical approval was obtained from the Provincial Research Ethics Committee of Granada (code 1585-N-20). Due to its observational nature, the aforementioned committee exempted the need to seek written informed consent. We utilized an anonymized database that was specifically designed for this study and identifying variables were removed for analysis.

## 3. Results

After excluding one patient aged < 18 years, there were 530 men (55%) and 438 women (45%) diagnosed with COVID-19 admitted to the participating hospitals. Their median age at admission was 67 years (range 18–100, interquartile range—IQR 55–77—but deceased subjects were notably older (Figure 1).

Over one in five patients were dependent in ADL. The majority (72%) of patients had at least one chronic condition, the most frequent of which was arterial hypertension (56%). Almost half of patients (43%) were polymedicated prior to admission. Median CURB-65 score upon admission was 1 (IQR 0–2), indicating low-severity, although patients who died presented with a significantly higher risk (median 2, IQR 2–3). Hydroxychloroquine (86%), and azithromycin (75%) were the most frequently prescribed drugs. During hospital stay, 117 patients (12%) were admitted to the ICU and 171 (18%) died. Patient characteristics are detailed in Table 1.

### 3.1. Patient Characteristics

Most patients were native (96%), aged 65 years or older (54%) and lived at home (86%). However, women were substantially more likely than men to be dependent in ADL (26% vs. 18%, *p* = 0.003) and to live in nursing homes (15% vs. 8%, *p* = 0.001). The most prevalent chronic conditions were arterial hypertension (56%), cardiovascular disease (25%), diabetes mellitus (23%), chronic lung disease (16%), chronic kidney disease (12%) and autoimmune disease (8%). Although their distribution was similar in men and women, cardiovascular disease (28% vs. 22%, *p* = 0.045) and COPD (11% vs. 2%, *p* < 0.001) were more frequent in men, whereas women suffered more frequently from autoimmune disease (10% vs. 6%, *p* = 0.028) and asthma (9% vs. 6%, *p* = 0.068). Deceased patients showed a significantly worse situation at baseline, regarding arterial hypertension, cardiovascular disease, chronic lung disease and chronic kidney disease (in all of them, *p* < 0.05 for both sexes). Among patients who died during hospital stay, the main positive associations between chronic conditions differed by sex: in men, (a) arterial hypertension with cardiovascular disease and (b) chronic kidney disease with active cancer; in women, the combination of diabetes with either (a) cardiovascular disease or (b) chronic kidney disease. Figure 2 shows all comorbidity correlations in deceased individuals.

At admission, most patients (88%) had an abnormal chest X-Ray and increased ferritin levels (median 478.2 µg/L), with deceased patients showing higher levels. Pneumonia severity was low in two thirds of patients, according to the CURB-65 score upon admission (median score of 2, IQR 0-2). Nevertheless, men who died showed a significantly higher score than those who survived (high risk in 34% vs. 4%, medium risk in 40% vs. 19%, low risk in 26% vs. 77%; *p* < 0.001). Even more flagrant differences were observed in women (high risk in 43% of deceased women vs. 3%, of discharged women, medium risk in 41% vs. 24%, low risk in 16% vs. 73%; *p* < 0.001). In total, 22% of patients developed concurrent infections, which were particularly frequent in deceased men (51%). Regarding treatments for COVID-19, 86% of the total cohort received hydroxychloroquine, 75% azithromycin, 65% other antibiotics (mainly third-generation cephalosporins) and 41% systemic corticosteroids; only a minority (12%) were given tocilizumab. Hydroxychloroquine, azithromycin and lopinavir-ritonavir were less frequently prescribed in patients who died. A relevant number of patients (114, 12%) received mechanical ventilation, with men needing IMV more often than women (11% vs. 5%, *p* = 0.005). With respect to the study outcomes, severe disease was more frequent in men, as they were twice as likely to be admitted to the ICU (15% vs. 8%, *p* = 0.001) and had a higher overall in-hospital mortality rate (19% vs. 16%, *p* = 0.244). Sex differences remained constant across age groups with the exception of those aged 90 years or older, of whom men had a lower in-hospital mortality (65% vs. 71%).

### 3.2. Survival Analysis

Figure 3 shows the survival curve of inpatients with COVID-19 during the first 30 days of follow-up, stratified by sex. The mortality rate was constant for men, whereas women died within the first 2 weeks, after which the mortality rate stabilized.

At day 7, the survival rate was 90.2% in men and 89.2% in women; at day 30, it was 68.3% in men and 78.08% in women. There were no significant differences between sexes (log-rank test *p* = 0.690). We also performed the survival analysis of men and women with COVID-19 stratified by CURB-65 score upon admission (Figure 4). Important differences were observed among patients with low risk (LR, score 0–1) and those with medium or high risk (MHR, score 2–5). Survival rate at day 30 in men was 80.1% in LR and 47.4% in MHR. In contrast, women with low risk kept an elevated survival rate throughout their hospital stay: at day 30, it was 93.4% in LR and 57.8% in MHR (log-rank test *p* < 0.001 for both sexes).

### 3.3. Multivariate Analysis

The multivariate Cox regression models (Table 2) identified five independent predictors of in-hospital mortality for each sex. In men, age hazard ratio—HR: 1.05 per additional year, 95% confidence interval—CI: 1.02–1.07), active cancer (HR: 2.78, 95% CI: 1.37–5.65), autoimmune disease (HR: 3.22, 95% CI: 1.55–6.69), CURB-65 score upon admission (HR:1.64 per additional point, 95% CI: 1.28–2.11) and azithromycin (HR: 0.53, 95% CI: 0.33–0.84) were associated with in-hospital mortality. In women, factors associated with in-hospital mortality were: age (HR: 1.06 per additional year, 95% CI: 1.02–1.09), cardiovascular disease (HR: 1.80, 95% CI: 1.02–3.18), chronic lung disease (HR: 1.84, 95% CI: 1.01–3.36), CURB-65 score upon admission (HR: 2.67 per additional point, 95% CI: 1.93–3.69) and azithromycin (HR: 0.50, 95% CI: 0.29–0.88). Additionally, we performed a third Cox regression model for all included patients (see Appendix A). The concordance index of the models was 0.84 (men), 0.90 (women) and 0.85 (both sexes).

## 4. Discussion

### 4.1. Predictors of In-Hospital Mortality

Characteristics of subjects included in this study resemble those reported in our country [16,17], so our cohort may be reasonably representative of the target population. The only remarkable differences were related to ICU admission: patients from our cohort were more likely to receive critical care, mechanical ventilation (IMV and NIMV) and tocilizumab. In-hospital mortality was somewhat lower than expected [18]: 19% in men and 16% in women. As regions where coronavirus struck earlier are usually overrepresented in large studies, variability in healthcare pressure and availability of ICU beds may underlie these disparities.

Advanced age is arguably the strongest risk factor for poor prognosis in COVID-19 [4]. In order not to lose information, we evaluated age as a continuous variable, finding a risk increase per each additional year of 5% in men and 6% in women, which is higher than most estimates reported [19]. A possible reason is the long average life expectancy in Andalusia, especially in women (85 years before the pandemic). Sex is also an independent risk factor for death [20]; a systematic review and meta-analysis estimated the chance of dying in males as 60% higher than women’s [21]. However, such value must be interpreted with caution as most studies included were Chinese and may not be generalizable to our study population. Men in our cohort showed a slightly higher risk than women (HR: 1.25, 95% CI: 0.88–1.77), but it cannot be concluded than male sex was associated with mortality. In a Spanish nationwide study investigating gender-based differences in over 10,000 hospitalised patients, men had a similar risk of death to those in our cohort (HR: 1.29, *p* < 0.001) [18]. A larger study examining the gender-related mortality of COVID-19 across European countries also found a higher risk in men (RR: 1.60, *p* < 0.001 [22]. Therefore, an insufficient statistical power might explain our results. A comprehensive discussion on differences among sexes is shown in Section 4.2.

Women present with milder symptoms at admission [18], which may be associated with disease outcomes. Although CURB-65 score upon admission was a strong predictor of in-hospital mortality in this study, its relevance differed strikingly between men and women (64% vs. 167% risk increase per 1-point increase, respectively). CURB-65, validated for community-acquired pneumonia (CAP) [23], has been widely used for predicting 30-day mortality in CAP but is far from ideal—PSI predicts mortality slightly better (91% vs. 88%) in patients with COVID-19 [24] and new scales have been developed for SARS-CoV-2 infection [25]. Yet, a study conducted in one of the participating hospitals [26] found that CURB-65 score had an adjusted HR of death of 1.76 per 1-point increase.

Azithromycin lowered the risk of death in men and women from our cohort. It was initially reported that this antibiotic reduced SARS-CoV-2 viral load, but unfortunately it did not show efficacy in clinical trials [27] and is no longer recommended to treat COVID-19. However, azithromycin might have improved the outcome of patients developing concurrent infection. To test this hypothesis, a subgroup analysis was performed, finding almost identical results in patients with and without concurrent infection. Beyond their antimicrobial activity, macrolides are used in infections such as pneumonia for their anti-inflammatory properties [28]. However, high-dose corticosteroids, which do have evidence supporting their use in severe COVID-19 [29], are a more powerful tool to counteract inflammation and were not associated with a reduction of in-hospital mortality. Neither of the effects of macrolides justify our findings but a potential selection bias might: azithromycin is more likely to be contraindicated in high-risk patients due to its side effects (prolongation of the QT interval, hepatotoxicity and multiple drug interactions).

Autoimmune disease and active cancer increased the risk of dying in men, in relation with an impaired immunity to infection. In women, cardiovascular disease and chronic lung disease were associated with in-hospital mortality. Cardiovascular morbidity hinders the prognosis of viral respiratory infections such as influenza [30], especially if thrombotic complications [31] occur. Respiratory chronic conditions, particularly COPD, worsen the outcomes of patients with COVID-19 by increasing viral susceptibility, disrupting ciliary clearance and compromising pulmonary function [32]. All of these predictors have been previously reported in Spain. For example, Poblador-Plou et al. [8] found that men with COVID-19 receiving immunosuppressive agents or corticosteroids for chronic conditions had a three-fold increase in the risk of death. However, most studies did not present their results stratified by sex: Berenguer et al. [17] observed a moderate increase in the hazard of death in patients with cancer; Muñoz-Rodríguez et al. [7] described similar results for the association between cardiac pathology and mortality; and Gude-Sampedro et al. [10] reported that risk of dying nearly doubled in patients with COPD. Of note, the low correlation we found between cancer and autoimmune disease in men, as well as that of cardiovascular disease and COPD in women suggests that specific associations of comorbidities may not be useful to predict mortality.

Notwithstanding that we included few clinical variables, the overall concordance (i.e., predictive accuracy) of the Cox regression models was high: in fact, its performance regarding mortality was better than that of prognostic models from regional studies in our country [10]. Even though complex scores based on clinical, analytical and radiological data are obviously more refined, predicting mortality using sociodemographic variables, chronic conditions and simple clinical information upon admission saves time—and time outweighs accuracy when health systems collapse. These findings reveal the practical implications of our work: to develop accurate risk scores adapted to the characteristics of patients in our region and to support the decision-making process of patient care when health care facilities are overwhelmed by the pandemic.

### 4.2. Sex and Gender Differences

Several sex-related biological factors influence COVID-19 susceptibility and severity. Two enzymes are crucial in the access of SARS-CoV-2 to host cells show sex-specific expression [33]. Transmembrane protease/serine subfamily member 2 (TMPRSS2) primes the viral spike protein to facilitate its binding to angiotensin-converting enzyme 2 (ACE2), which acts as the entry receptor. The density of ACE2 receptors in each tissue correlates to the damage inflicted by the virus, with the lower respiratory tract (type II pneumocytes) constituting the main target [34].

The immune defence against SARS-CoV-2 appears to be of paramount importance. Innate and adaptive immune response are more robust in females [35]. Activation of antigen-presenting cells by toll-like receptors (TLRs) is a key element of innate immunity; coronavirus can interfere with TLR7 signalling, but higher expression of TLR7 in females reduces the inhibitory effect of the virus [36]. An excess of high-mobility group box 1 protein (HMGB1) induces release of interleukin-6 [37], a pro-inflammatory agent with a protagonist role in the cytokine storm [38]. Differentially regulated among sexes, males show higher levels of HMGB1 [39]. Another biomarker of disease severity, lymphopenia, is also more frequent in male patients [40]. Furthermore, females produce higher titres of serum IgG [41], which may help them control severe infection. In fact, male sex is one of the factors behind prolonged viral shedding [42], despite the absence of sex differences in viral load [43]. Altogether, women would possess better tools to tackle SARS-CoV-2.

In addition, some alternative explanations have been proposed. Sex hormones modulate cellular and humoral immunity responses against infection [44]. Oestrogens are immune-stimulatory [45], while testosterone is immunosuppressive and may lead to thrombotic complications by inducing platelet activation and aggregation [46]. Nonetheless, it plays a cardioprotective role in men [20]. Otherwise, females would have higher endurance levels under stressful conditions [39]. Last, intestinal microbiota regulates the testosterone levels and is affected by oestrogens [45]. Thus, the possible interaction between SARS-CoV-2 and gut commensal bacteria [47] could be sex-biased.

Less attention has been drawn to psychosocial factors, such as gender, which is a social determinant of health. Women live longer but have fewer healthy years, experiencing pain [48], certain diseases [49] and mental health problems [50] more often than men. Women are also more vulnerable during epidemics for a variety of reasons: some risk factors for mental health disorders (e.g., social isolation) are disproportionately frequent in women [51]; the gender pay gap [52] and the barriers in the labour market compromise women’s financial security in times of crisis; publications and public health documents usually fail to address gender issues [53], for instance, by only taking into account one modality of care (formal healthcare) [54]; access to specific treatments is delayed as women are underrepresented in drug development research [55], and those pregnant or breastfeeding are excluded from most clinical trials [56]. To further complicate matters, epidemics exacerbate gender inequities—women take on the functions that the system can no longer assume [57]—and the ongoing pandemic is not an exception [58].

Infection by SARS-CoV-2 is more frequent in women [59], perhaps because of a higher degree of exposure to the virus, particularly in the first wave of the pandemic, when shortage of protective equipment was common across countries [60]. Women account for 68% of healthcare professionals and 76% of the providers of informal care in Spain [61]. In contrast, differences in behavioural factors are detrimental to men, who wait longer before seeking care [62] and consume addictive substances more frequently. Three in five smokers in Spain are men [63], and tobacco use is correlated with COVID-19 prognosis. Additionally, self-reported adherence to public health measures is lower in men [64].

The first wave of the pandemic witnessed gender-based consequences of coronavirus disease and the implementation of lockdowns. Social disruption increased the risk of suicide [65]; furthermore, it has been suggested that women particularly struggled to maintain their mental wellbeing [51], which should not come as a surprise, given the unique stress, fear and guilt that women carry during epidemics [66]. It is also worth mentioning that lockdowns had indirect health effects on women, including: a considerably higher rate of caesarean delivery [67], a lack of access to abortion [68] and a concerning increase in gender-based violence [69].

As with mortality, identifying which factors account for differences in COVID-19 outcomes can be extraordinarily challenging. In the ANCOHVID cohort, systemic (fatigue and musculoskeletal pain) and mental health sequelae (depressive and anxiety symptoms) after hospitalisation were significantly more frequent in women, whereas uncontrolled glycaemia was more frequent in men [14]. Who is to blame, sex or gender? Probably both, as they are closely intertwined [70,71]: without embracing the interrelation between biological and social factors, it will be hard to fathom the differential impact of coronavirus among men and women.

### 4.3. Strengths and Limitations

Our study presents several limitations. There are some weaknesses inherent to its retrospective nature, such as the quality and completeness of information based on clinical records. Obesity and smoking are associated with a poor prognosis of COVID-19; yet, they could not be analysed in this study given the lack of data collected in most patients. In our opinion, behavioural risk factors should be properly collected in clinical settings, since they influence health and disease similarly to biological risk factors. Besides, nursing homes were worst hit in the first wave of the pandemic in Spain [72]. Thousands of institutionalized patients died in these facilities due to strict hospital admission criteria [73]. Thus, there is a risk of selection bias that would underestimate mortality. However, we conducted a 6-month follow-up until October 2020 and believe that these findings are useful for current literature, due to the paucity of data on sex differences and the course of the pandemic. Finally, there is also a risk of confusion bias, since we did not adjust for relevant risk factors with missing data (e.g., obesity) or that were not collected (e.g., cerebrovascular disease). However, we designed a study with a high external validity, as it is multi-centred and has a large sample size; and a robust analysis of sex differences in patients hospitalised with COVID-19.

## 5. Conclusions

We described the baseline characteristics, treatments received and main outcomes of patients with COVID-19 admitted to four centres in Andalusia, Spain, in the first wave of the pandemic. We analysed predictors of in-hospital mortality. In both sexes, advanced age, higher CURB-65 score upon admission and not receiving treatment with azithromycin increased the risk of death. Sex-specific risk factors were active cancer and autoimmune disease in men, and cardiovascular disease and chronic lung disease in women.

In this study, men show higher rates of ICU admission and in-hospital mortality. Nevertheless, this may not be the case in middle- and especially low-income countries, where less research is conducted. Another topic that should be addressed in future studies is the long-term impact of the pandemic, as it might be greater on women. Several authors [58,74] have made a call to address the sex- and gender-specific long-run effects of COVID-19 on population health: healthcare systems are facing the growing demand for care and the mental health fallout [51]. To this end, governments must allocate resources [75] to overcome barriers to health services access and expand social protection.

## Figures and Tables

**Figure 1 ijerph-18-09018-f001:**
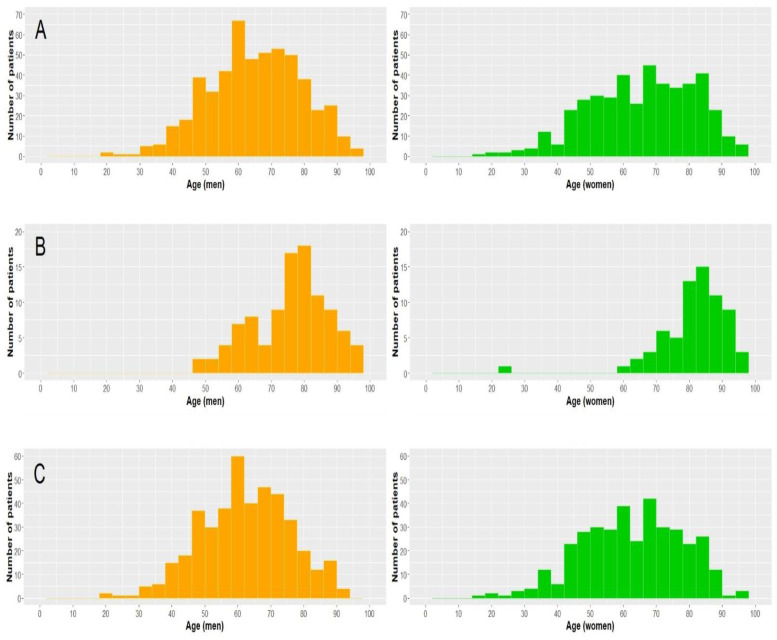
Distribution of all admitted patients (**A**), patients who died (**B**) and patients who survived (**C**), stratified by age and sex.

**Figure 2 ijerph-18-09018-f002:**
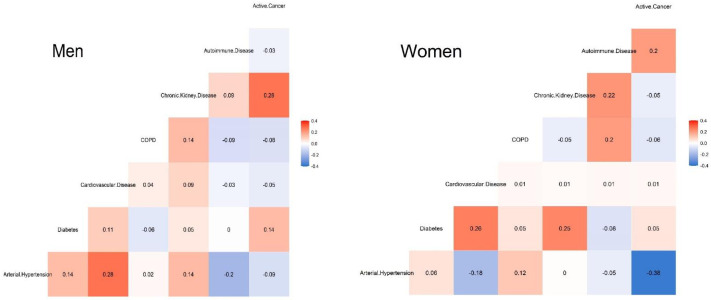
Correlation matrices of comorbidities in men (**left**) and women (**right**) in patients hospitalised with COVID-19 who died during hospital stay.

**Figure 3 ijerph-18-09018-f003:**
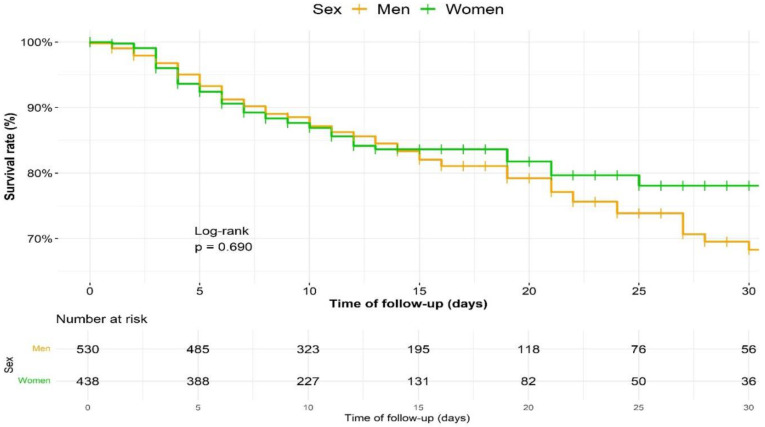
Survival analysis of men and women, until day 30 of hospitalisation.

**Figure 4 ijerph-18-09018-f004:**
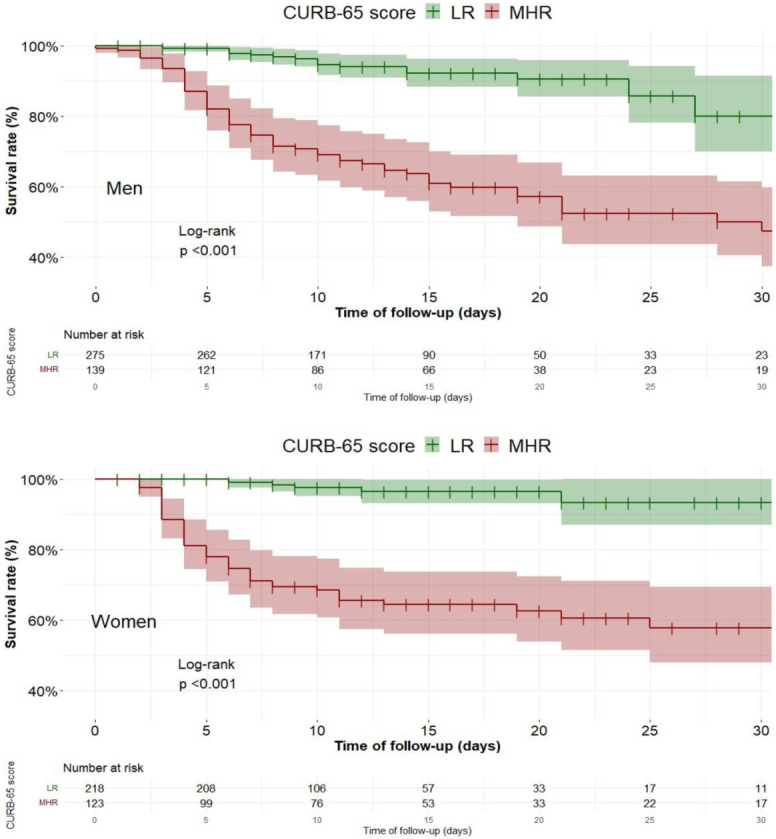
Survival analysis of men (**top**) and women (**bottom**), stratified by low (LR) or medium/high risk (MHR) according to CURB-65 score upon admission, until day 30 of hospitalisation.

**Table 1 ijerph-18-09018-t001:** Characteristics of patients hospitalised with COVID-19, stratified by sex.

Characteristic	Total (*n* = 968)	Men, Dead (*n* = 101)	Men, Alive (*n* = 429)	*p* Value (men) ^a^	Women, Dead (*n* = 70)	Women, Alive (*n* = 368)	*p* Value (women) ^a^
**Age**, years				<0.001			< 0.001
<40	44 (5.5%)	0 (0.0%)	19 (4.4%)		1 (1.4%)	25 (6.8%)	
40–49	93 (11.7%)	2 (2.0%)	50 (11.7%)		0 (0.0%)	43 (11.7%)	
50–59	181 (22.7%)	8 (7.9%)	100 (23.3%)		0 (0.0%)	81 (22.0%)	
60–69	207 (26.0%)	16 (15.8%)	120 (28.0%)		6 (8.6%)	88 (23.9%)	
70–79	164 (20.6%)	30 (29.7%)	94 (21.9%)		15 (21.4%)	69 (18.8%)	
80–89	95 (11.9%)	32 (31.7%)	39 (9.1%)		33 (47.1%)	56 (15.2%)	
≥90	13 (1.6%)	13 (12.9%)	7 (1.6%)		15 (21.4%)	6 (1.6%)	
Mean (IQR)	67 (55–77)	77 (68–84)	63 (54–72)		83 (77.5–88)	63 (52–75)	
**Country of birth**				0.509			0.392
Non-native	36 (3.9%)	1 (1.0%)	34 (4.4%)		1 (1.5%)	19 (5.4%)	
**Centre**				0.257			0.692
Granada (SCUH)	441 (45.6%)	52 (51.5%)	191 (44.5%)		28 (40.0%)	170 (46.2%)	
Jaén (CJUH)	270 (45.6%)	26 (25.7%)	125 (29.1%)		20 (28.6%)	99 (26.9%)	
Córdoba (RSUH)	220 (22.7%)	16 (15.8%)	95 (22.1%)		19 (27.1%)	90 (24.5%)	
Cádiz (PRUH)	37 (3.8%)	7 (6.9%)	18 (4.2%)		3 (4.3%)	9 (2.5%)	
**Dependence in activities of daily living**	207 (21.5%)	41 (41.0%)	53 (12.4%)	<0.001	47 (68.1%)	66 (18.0%)	<0.001
**Place of residence**							
Living at home	826 (85.6%)	72 (72.7%)	393 (91.8%)	<0.001	42 (60.0%)	319 (86.7%)	<0.001
Nursing homes	104 (10.9%)	24 (24.2%)	17 (4.0%)	<0.001	26 (37.7%)	37 (11.5%)	<0.001
Institutions for disabled people	39 (4.1%)	3 (3.1%)	20 (4.7%)	0.594	3 (4.3%)	13 (3.6%)	0.731
*Missing data*	33 (3.4%)						
**Chronic conditions**							
No. of chronic conditions; median (IQR)	1 (0–2)	2 (1–4)	1 (0–2)	<0.001	2 (1–4)	1 (0–2)	<0.001
Arterial hypertension	542 (56.0%)	75 (74.3%)	226 (52.7%)	<0.001	56 (80.0%)	185 (50.3%)	<0.001
Diabetes mellitus	226 (23.3%)	32 (31.7%)	95 (22.1%)	0.059	27 (38.6%9	72 (19.6%)	<0.001
Cardiovascular disease	243 (25.1%)	46 (45.5%)	101 (23.5%)	<0.001	32 (45.7%)	64 (17.4%)	<0.001
Chronic lung disease	154 (15.9%)	29 (28.7%)	62 (14.5%)	0.001	19 (27.1%)	44 (12.0%)	0.002
COPD	65 (6.7%)	21 (20.8%)	36 (8.4%)	<0.001	4 (5.7%)	4 (1.1%)	0.025
Asthma	69 (7.1%)	6 (5.9%)	24 (5.6%)	1.000	4 (5.7%)	35 (9.5%)	0.428
Chronic kidney disease	112 (11.6%)	21 (20.8%)	41 (9.6%)	0.003	23 (32.9%)	27 (7.3%)	<0.001
Autoimmune disease	74 (7.6%)	9 (8.9%)	22 (5.1%)	0.222	4 (5.7%)	39 (10.6%)	0.299
Immunosuppression	41 (4.2%)	4 (4.0%)	16 (3.7%)	1.000	7 (10.0%)	14 (3.8%)	0.059
Polymedication (≥6 drugs prior to admission)	403 (42.6%)	59 (59.6%)	152 (36.3%)	<0.001	49 (72.1%)	143 (39.7%)	<0.001
*Missing data*	22 (2.3%)						
Active cancer	50 (5.2%)	14 (13.9%)	20 (4.7%)	0.002	4 (5.7%)	12 (3.3%)	0.301
History of cancer in the previous 5 years	62 (6.4%)	9 (8.9%)	35 (8.2%)	0.963	6 (8.6%)	12 (3.3%)	0.051
Solid organ or HSC transplantation	10 (1.0%)	0 (0.0%)	6 (1.4%)	0.601	2 (2.9%)	2 (0.5%)	0.122
**In-hospital variables**							
Length of stay (days); median (IQR)	11 (7–17)	8 (4–15)	12 (8–18)	<0.001	6 (4–11)	10 (7–17)	<0.001
Length of ICU stay (days); median (IQR)	12 (6–3.25)	13 (9–26)	13 (5.5–30)	0.533	10 (6–13)	12 (4.5–15)	0.840
Abnormal admission chest X-ray	801 (87.8%)	80 (89.9%)	373 (90.3%)	0.941	53 (85.5%)	295 (84.8%)	0.962
Ferritin upon admission (µg/L); median (IQR)	478.2 (246.8–866.7)	732.8 (453.4–1229.9)	654.0 (393.1–1093.8)	0.109	349.6 (149.0–712.8)	277.2 (132.7–505.5)	0.123
CURB-65 score upon admission; median (IQR)	1 (0–2)	2 (1–3)	1 (0–1)	<0.001	2 (2–3)	1 (0–2)	<0.001
Low risk (CURB-65 = 0–1)	493 (63.3%)	22 (25.9%)	252 (76.9%)		9 (16.1%)	209 (73.3%)	
Medium risk (CURB-65 = 2)	186 (24.6%)	34 (40.0%)	62 (18.8%)		23 (41.1%)	67 (23.5%)	
High risk (CURB-65 = 3–5)	76 (10.1%)	29 (34.1%)	14 (4.3%)		24 (42.9%)	9 (3.2%)	
*Missing data*	213 (22.0%)						
Concurrent infection	166 (22.0%)	41 (51.3%)	66 (20.2%)	<0.001	13 (25.0%)	46 (15.6%)	0.143
*Missing data*	214 (22.1%)						
Hydroxychloroquine	804 (86.3%)	61 (64.2%)	381 (92.3%)	<0.001	39 (58.2%)	323 (90.5%)	<0.001
*Missing data*	36 (3.7%)						
High-dose corticosteroids	362 (41.1%)	50 (55.6%)	182 (46.4%)	0.148	27 (42.2%)	103 (30.8%)	0.100
*Missing data*	87 (9.0%)						
Lopinavir-ritonavir	569 (62.0%)	54 (56.8%)	278 (68.0%)	0.052	26 (39.4%)	211 (60.6%)	0.002
*Missing data*	50 (5.2%)						
Azithromycin	680 (74.6%)	48 (53.3%)	317 (77.9%)	<0.001	30 (47.6%)	285 (81.0%)	<0.001
*Missing data*	56 (5.8%)						
Other antibiotics	581 (65.1%)	63 (70.8%)	268 (67.9%)	0.680	41 (64.1%)	209 (60.8%)	0.720
*Missing data*	76 (7.9%)						
Tocilizumab	100 (11.8%)	14 (16.1%)	58 (15.6%)	0.982	4 (6.8%)	24 (7.3%)	0.890
*Missing data*	120 (12.4%)						
Invasive mechanical ventilation	81 (8.4%)	23 (22.8%)	34 (7.9%)	<0.001	7 (10.0%)	17 (4.6%)	0.084
Non-invasive mechanical ventilation	88 (9.1%)	11 (10.9%)	49 (11.4%)	1.000	4 (5.7%)	24 (6.5%)	1.000
ICU admission	117 (12.1%)	25 (24.8%)	56 (13.1%)	0.005	9 (12.9%)	27 (7.3%)	0.192

CJUH, City of Jaén University Hospital; COPD, chronic obstructive pulmonary disease; CURB-65, prognostic scale based on blood urea nitrogen, respiratory rate, blood pressure and age; HSC, hematopoietic stem-cell; ICU, intensive care unit; IQR, interquartile range; PRUH, Puerto Real University Hospital; RSUH, Reina Sofía University Hospital; SCUH, San Cecilio University Hospital. Percentage of variables with missing data are reported. ^a^ *p*-value of Mann–Whitney U test, chi-square test, or Fisher’s exact test, when appropriate.

**Table 2 ijerph-18-09018-t002:** Cox regression models for in-hospital mortality among men and women.

Predictors	Crude HR ^a^ (95% CI) ^b^	*p* Value ^c^	Adjusted HR (95% CI)	*p* Value ^c^
**Men**				
Age (years)	1.08 (1.06–1.10)	<0.001	1.05 (1.02–1.07)	<0.001
Active cancer	2.26 (1.28–3.98)	0.005	2.78 (1.37–5.65)	0.005
Autoimmune disease	1.73 (0.87–3.43)	0.119	3.22 (1.55–6.69)	0.002
CURB-65 score	2.32 (1.88–2.86)	<0.001	1.64 (1.28–2.11)	<0.001
Azithromycin treatment	0.38 (0.61–2.30)	<0.001	0.53 (0.33–0.84)	0.008
**Women**				
Age (years)	1.09 (1.07–1.12)	<0.001	1.06 (1.02–1.09)	0.002
Cardiovascular disease	3.00 (1.86–4.82)	<0.001	1.80 (1.02–3.18)	0.044
Chronic lung disease	1.76 (1.02–3.02)	0.042	1.84 (1.01–3.36)	0.045
CURB-65 score	3.31 (2.54–4.32)	<0.001	2.67 (1.93–3.69)	<0.001
Azithromycin treatment	0.24 (0.15–0.40)	<0.001	0.50 (0.29–0.88)	0.016

^a^ Hazard ratio; ^b^ Confidence interval; ^c^ *p*-value of Wald’s test. The concordance index of the models was 0.84 for men and 0.90 for women.

## Data Availability

Data presented in this study are available on request from the corresponding author.

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
