# Peer review of "Sex Differences and Predictors of In-Hospital Mortality among Patients with COVID-19: Results from the ANCOHVID Multicentre Study"

_ijerph, 2021, doi:10.3390/ijerph18179018_

Round 1
Reviewer 1 Report
It is a well-crafted manuscript that describes a well-executed work with an appropriate design, on a relevant topic.
There are a few comments that the authors might respond to:
Introduction section (page 2, line 70): it should be indicated to what date these data correspond.
Material and Methods Section (page 2, line 92): the document (and the date) from which the case definition is taken should be cited.
Discussion section (page 14, line 462): perhaps it would be interesting to highlight the lack of data completion in precisely relevant risk factors (obesity and smoking), as cited in the manuscript, which are two of the three variables that They have not been used (due to missing or uncollected data), and taking advantage of this fact to reinforce the importance that should be given in the data collected in the hospital setting to these types of factors, they are often considered less interesting in the clinical setting.
Author Response
Dear reviewer,
Thank you very much for your comments and suggestions. We hope that we have been able to answer all your concerns. The point-by-point response to all the suggestions can be found in this document. You will find the changes regarding your revision in the main text of the reuploaded manuscript (using the track-changes function), highlighted in yellow coloured text.
It is a well-crafted manuscript that describes a well-executed work with an appropriate design, on a relevant topic. There are a few comments that the authors might respond to:
Introduction section (page 2, line 70): it should be indicated to what date these data correspond.
These data correspond to May 10, 2020 (the end of the first wave of the pandemic in Andalusia, Spain).
This information is now included in the Introduction section (page 2, line 71).
Material and Methods Section (page 2, line 92): the document (and the date) from which the case definition is taken should be cited.
The case definition was taken from the Spanish Ministry of Health Strategy for Early Detection, Surveillance and Control of COVID-19 version at the beginning of the inclusion period (March 1 2020), which was then updated four times until the last version (April 10 2020). The case definition changed once during the inclusion period: a positive PCR test to SARS-CoV-2 was required in all versions of the Ministry of Health Strategy except the last one, which also included patients with a positive antigen test result or with a positive serological test result as confirmed cases. However, for this study, we only included patients with a confirmed polymerase chain reaction (PCR) to SARS-CoV-2.
The first document and its creation date have been included in the Introduction section (page 2, lines 96 and 97), with reference No. #15.
Discussion section (page 14, line 462): perhaps it would be interesting to highlight the lack of data completion in precisely relevant risk factors (obesity and smoking), as cited in the manuscript, which are two of the three variables that they have not been used (due to missing or uncollected data), and taking advantage of this fact to reinforce the importance that should be given in the data collected in the hospital setting to these types of factors, they are often considered less interesting in the clinical setting.
We truly appreciate the advice of the reviewer on this point. Behavioural risk factors related to life habits are sometimes poorly collected in clinical settings and therefore hinder the study of their contribution to disease outcomes. We have highlighted the importance of appropriate collection of this type of data in the Discussion section (page 14, lines 467 to 470).
Reviewer 2 Report
This study demonstrates sex differences and predictors of in-hospital mortality among patients with COVID-19. Revision is required for the following points.
First, authors should added extensive reviews about previous studies oo gender difference of mortality.
Second, add previous studies of control variables (mainly underlying diseases) and compare them with the results of this study.
Third, please suggest what practical implications this study can give.
Fourth, are there any statistics related to the demographic characteristics of the area where the data were collected? It is likely that basic data related to life expectancy and health infrastructure should be added.
Fifth, authors provide what topics require future research on gender difference.
Author Response
Dear reviewer,
Thank you for all your comments and suggestions. We hope that we successfully answered all your concerns and improved the overall quality of the manuscript. It can be found in this document the point-by-point response to all the suggestions and, in the main text of the reuploaded manuscript (using the track-changes function), you will find the changes regarding your revision in green coloured text.
This study demonstrates sex differences and predictors of in-hospital mortality among patients with COVID-19. Revision is required for the following points.
First, authors should added extensive reviews about previous studies oo gender difference of mortality.
Previous studies about gender differences in COVID-19 mortality were reviewed. The Discussion section (page 11, lines 338 to 349) underlines the two main systematic reviews and meta-analyses on this matter (references No. #20 and #21) and compares our findings with that of the largest study from our country (reference No. #18). Additionally, sex and gender differences are later discussed in depth (pages 12 and 13, lines 400 to 463). However, according to the reviewer’s suggestion, we have further reviewed the literature on this topic. Therefore, the discussion section now also includes a study across multiple European countries (page 11, lines 345 to 347, with reference No. #22.
Second, add previous studies of control variables (mainly underlying diseases) and compare them with the results of this study.
We appreciate the suggestion. We reviewed previous studies of control variables. Underlying diseases are discussed in the Discussion section (page 12, lines 378 to 386): cancer (reference No. #17), cardiovascular disease (reference No. #7), lung disease (reference No. #10) and autoimmune disease (reference No. #8).
Control variables different from underlying diseases were also reviewed in the Discussion section (pages 11 and 12, lines 333 to 371): advanced age (reference No. #4 and #19), CURB-65 score at admission (reference No. #24 and #26), azithromycin treatment (reference No. #27), and sex (previously discussed).
Third, please suggest what practical implications this study can give.
We agree with the reviewer that the practical implications of this study should be included in the manuscript. We believe that our findings can help develop prognostic scores tailored to patients from our regions and support the decision-making process in situations of high healthcare pressure.
Consequently, this information is now available in the Discussion section (page 12, lines 395 to 398).
Reviewer 3 Report
Thank you for your submission. Data analyses of the paper were sound. However, the patient data from March 1, 2020 to April 15, 2020, seems to to be out-of-date, and information from results of this study also seems to be out-of-date. There have been so much information from many new epidemiological studies and drug trials published from April 16, 2020 to present. Thus we can accept your paper if you will do same kind of analyses for more recent patient data. Thanks again.
Author Response
Dear reviewer,
Thank you for your suggestions. We completely understand your concerns about the study period. As much as we would like to update the analyses, we do not have access to more recent patient data because the last patient included in our cohort was admitted on April 15, 2020. However, data analysed in the study encompasses the period until October 15 2020, as every patient was followed for six months after the date of admission if they were still hospitalised. This information has been included in the Materials and Methods section (page 2, line 86 and 87) using the track-changes function, in blue coloured text.
Thank you for your submission. Data analyses of the paper were sound. However, the patient data from March 1, 2020 to April 15, 2020, seems to to be out-of-date, and information from results of this study also seems to be out-of-date. There have been so much information from many new epidemiological studies and drug trials published from April 16, 2020 to present. Thus we can accept your paper if you will do same kind of analyses for more recent patient data. Thanks again.
We do not deem our results to be out-of-date for the following reasons. First, vaccines against COVID-19 have improved but not ended the pandemic, since their effectiveness is not perfect, particularly in a scenario of global vaccine inequity and the rise of more transmissible SARS-CoV-2 variants. Second, healthcare systems are still collapsing as the availability of beds is limited – the situation has become dramatic in countries such as India and Indonesia, which have recently recorded a huge COVID-19 death toll. Nevertheless, even in countries with high vaccination rates like Spain, several regions have reached their maximum capacity for intensive care during the fifth wave. Third, sex and gender differences keep being disregarded, and most of the health policies necessary for addressing these differences have not been taken yet in our country. Finally, we used a multicentre cohort that has already been completed and truly believe that these data are valuable for current literature and future research.
We hope that we have provided arguments to justify why our data remain in place. Acknowledgement and justification of the study period have been included in the Discussion section (page 14, lines 473 to 476).
Round 2
Reviewer 2 Report
Authors revised all of things commented by reviewers
Reviewer 3 Report
Thank you for showing us the follow up duration until October 2020.